# Metabolic and bariatric surgery among patients with social anxiety disorder, a matched cohort study

Jesper Christiansen[1], Erik Näslund[2], Henrik Larsson[3,4], Erik Stenberg[1]*

1 Department of Surgery, Faculty of Medicine and Health, Örebro University, Örebro, Sweden, 2 Division of Surgery, Department of Clinical Sciences, Danderyd Hospital, Karolinska Institutet, Stockholm, Sweden, 3 School of Medical Sciences, Örebro University, Örebro, Sweden, 4 Department of Medical Epidemiology and Biostatistics, Karolinska Institutet, Stockholm, Sweden

* erik.stenberg@oru.se, erik.stenberg@regionorebrolan.se

## Abstract

Social anxiety disorder is common among patients considered for metabolic and bariatric surgery (MBS). The combination of social anxiety with obesity may, however, be associated with a higher risk for adverse outcomes after surgery. In this nationwide, registry-based, matched cohort study, all patients who underwent primary MBS in Sweden from 2007 until 2019 and who had a diagnosis of moderate to severe social anxiety disorder (n = 586) were matched using a Propensity score to controls who underwent the same treatment but who did not have social anxiety disorder (n = 5791) with a mean follow-up time of 6.9 years. Patients with social anxiety disorder experienced an increased risk for non-serious postoperative complications (OR 1.59; 95%CI 1.21–2.09), self-harm (HR 2.44 CI 95% 1.84–3.25 p < 0.001) and alcohol or substance abuse (HR 2.41, 95%CI 1.96–2.96, p < 0.001), and reported lower psychosocial health-related quality of life before and after surgery. However, patients with social anxiety disorder significantly improved in health-related quality of life compared to baseline, and experienced similar effects on weight reduction at 2 years after surgery (total weight loss: 32.8 ± 10.3% compared to 32.6 ± 9.7%) and risks for cardiovascular events compared to the matched control group. MBS appears to be a safe and effective treatment for severe obesity in patients with social anxiety disorder, but an individualized and increased peri- and postoperative support should be considered for patients with moderate to severe social anxiety disorder and severe obesity.

## Introduction

The global prevalence of obesity is rising, reaching pandemic levels. Living with severe obesity is associated with an increased risk of metabolic comorbidities, including type-2 diabetes, cardiovascular disease, and certain cancers [1]. Additionally,

**Data availability statement:** Data cannot be shared publicly because of patient confidentiality under current Swedish legislation. Data are available for researchers who meet the criteria for access to confidential data (contact via the Scandinavian Obesity Surgery Registry; soreg@regionorebrolan.se).

**Funding:** This work was supported by grants from Region Örebro County, Åke Wiberg Foundation, Stockholm County Council, and SRP Diabetes. The funders had no role in study design, data collection and analysis, decision to publish, or preparation of the manuscript.

**Competing interests:** Henrik Larsson reports receiving grants from Shire Pharmaceuticals; personal fees from and serving as a speaker for Medice, Shire/Takeda Pharmaceuticals and Evolan Pharma AB; all outside the submitted work. HL is editor-in-chief of JCPP Advances. Erik Stenberg have received reimbursement for lecture from MSD and consultant fees from Johnson & Johnson Medical (to institution) for work unrelated to the submitted work. None of the remaining authors declare any conflict of interest.

depression and anxiety disorders – which are already common in the general population – have a bidirectional relationship with obesity [2].

Social anxiety disorder is a specific form of anxiety characterized by an intense fear of humiliation or rejection during public performance or in social interactions. In moderate to severe cases, this fear can lead to pronounced social avoidance and isolation. It affects an estimated 5–10% of the general population worldwide, but prevalence rates can be as high as 18% among candidates for metabolic and bariatric surgery (MBS) [3,4]. Social anxiety disorder may negatively impact engagement in important daily activities, reduce adherence to postoperative follow-up, and has been identified as a risk factor for non-compliance with nutritional supplementation following surgery [5,6].

While MBS has demonstrated significant benefits for many metabolic comorbidities [7], its effects on mental health-related quality of life are more limited, with minimal long-term improvements in social functioning [8,9]. Moreover, there is evidence of an increased risk of depression, alcohol misuse, and self-harm after MBS [10]. Postoperative outcomes in patients with attention deficit hyperactivity disorder (ADHD) have been shown to be influenced by follow-up adherence [11], suggesting that coexisting social anxiety may also complicate recovery and long-term success after MBS in patients living with severe obesity.

Despite the high prevalence of social anxiety disorder among patients being evaluated for MBS, there is a lack of research on surgical outcomes in those with moderate to severe forms of the disorder. The present study, therefore, aims to evaluate the safety and efficacy of MBS in patients with moderate to severe social anxiety disorder.

## Methods

This study was conducted using record linkage of the Scandinavian Obesity Surgery Registry (SOReg) with nationwide Swedish health registers, using the unique personal identity number assigned to each Swedish resident. SOReg is a national quality registry reporting preoperative, intraoperative, and follow-up data 6 weeks, 1, 2, 5, and 10 years after surgery. The registry covers virtually all MBS procedures in Sweden at present, and has been shown to have a very high acquisition and internal validity [12]. The cross linkage included the Swedish Prescribed Drug Register, established in 2005, including all dispensed prescription drugs classified according to the World Health Organization Anatomical Therapeutic Chemical (ATC) classification system, the mandatory National Patient Register (NPR), containing valid inpatient and outpatient hospital care data since 1987 [13], the Total Population Register, continually updated by Statistics Sweden, provided data on emigration/immigration, and dates of birth/death [14], and the Longitudinal Integrated database for health Insurance and Labour market studies (LISA) provided by Statistics Sweden provided socioeconomic data for all individuals [15]. All data were available at an individual level, but personal identification numbers were replaced with pseudonymized identification using a code key that was only available to the national authorities (Statistics

Sweden and the National Board of Health and Welfare). The data were made available for the researchers on 17 April 2023.

## Inclusion and exclusion criteria

In accordance with current indications for MBS, adults (≥18 years of age) with a BMI ≥ 35 kg/m$^2$ or ≥30 kg/m$^2$ with meta-bolic comorbidity, who underwent non-revisional, primary Roux-en-Y gastric bypass (RYGB) or sleeve gastrectomy (SG), between 2007 and 2017, were considered for inclusion [16].

## Study population and intervention

Social anxiety disorder was defined as a diagnosis of social anxiety disorder (ICD-10. F40.1) based on registration during inpatient care or specialist outpatient care in the NPR [17].

Based on a Power calculation with an assumption of a difference in percentage total weight loss (%TWL) from the national median weight loss of 28.3 to 26.9 reported for patients with depression, would require 537 operated with social anxiety disorder and 5370 operated controls without social anxiety disorder [18]. The propensity score matching (1:10) was stratified by surgical method and included (nearest function) sex, age, body mass index (BMI), sleep apnea, hyper-tension, type-2 diabetes (T2D), dyslipidemia, chronic obstructive pulmonary disease (COPD), cardiovascular disorder, disposable income, unemployment, origin, year of surgery, surgical method, access, and surgical center.

The surgical technique for the laparoscopic RYGB was highly standardized during the study period with an antecolic, antegastric RYGB with a small gastric pouch (<25 mL), an alimentary limb of 100 cm, and a biliopancreatic limb of 50 cm [19]. The surgical technique for the laparoscopic SG was less standardized, but routinely performed using a 32–36 French bougie, starting the resection ≤5 cm from the pylorus, ending the resection 1 cm from the angle of His.

## Covariates

Age, sex, disposable income, and origin were based on individual data from the Total Population Register and Statistics Sweden. Disposable income (total taxable income minus taxes and other negative transfers) was indexed to the 2019 consumer price index and divided into quartiles based on the indexed disposable incomes of all patients operated on with MBS in Sweden. Origin was divided into three categories based on country of birth and parents' country of birth.

Baseline BMI, and the presence of sleep apnea, depression, T2D, dyslipidemia, and hypertension were based on data from the SOReg and defined as a condition receiving active treatment (continuous positive airway pressure and pharmacological treatment, respectively) at the time of surgery. Previous substance abuse, COPD, and cardiovascular comorbidity were based on combined data from the SOReg, the National Patient registers, and the Prescribed Drugs register. Cardiovascular comorbidity was defined as a previous diagnosis of heart failure (ICD-10: I50), acute myo-cardial infarction or angina pectoris (ICD-10: I20-22), atrial fibrillation, flutter, or other tachycardia (ICD-10: I47-48). COPD was defined as hospital admission for COPD or a complication of COPD with COPD as secondary diagnosis in the NPR for in-hospital care (ICD-10: J44) or a prescription of an anticolinergic drug (ATC-code:R03BB), long-acting beta-2 antagonist (ATC-codes: R03AC12-R03AC18), or a combination of these (ATC-code: R03AL) indicating moder-ate to severe COPD [20]. Alcohol or substance abuse was defined as previous hospital admission or outpatient care at a specialist clinic for substance abuse (ICD-10: F10-F16, or prescription of ATC-code: N07BB at any time before surgery).

## Outcome and follow-up

Outcome measures were early postoperative complications (occurring within 30 days of surgery), postoperative follow-up attendance, weight change from baseline (before preoperative weight-reduction) to the follow-up at 2 years after surgery,

health-related quality-of-life (HRQoL), major adverse cardiovascular event (MACE), and late complications (self-harm and alcohol or substance abuse) as well as overall mortality. Early postoperative complications were defined as specific complications requiring prolonged hospital stay, readmission, or intervention. A serious postoperative complication was defined as a complication requiring intervention under general anesthesia, resulting in organ failure or death (≥IIIb on the Clavien-Dindo scale) [21], with information available for patients who underwent surgery from January 1, 2010 (the Clavien-Dindo classification was introduced in SOReg on January 1, 2010).

HRQoL was assessed using the Obesity-related problems scale, which is a disease-specific instrument measuring the impact of obesity on psychosocial functioning. The scale reports a summary score from 0 to 100, with a higher score indicating more psychosocial dysfunction [22]. MACE was defined as the first occurrence of unstable angina (ICD-10: I20.0), acute myocardial infarction (ICD-10: I21-22), cerebrovascular event (ICD-10: I60,61,63 or 64), fatal cardiovascular event (cause of death ICD-10: 101−78, excluding I30) or unattended sudden cardiac death (ICD-10: R96.0,R96.1,R98 and R99). Self-harm was defined as the first admission or treatment for self-inflicted serious injury or intoxication (ICD-10: X60-84), or a cause of death caused by self-induced injury (ICD-10: X60-84) or injury of unclear intent (ICD-10: Y10-34). Alcohol or substance abuse was defined as hospital admission or visit to a specialist clinic for alcohol or substance abuse (ICD-10: F10-F16), or a prescription of drugs for alcohol abuse (ATC-code: N07BB).

Participants were followed surgery until emigration, death, or end of follow-up (December 31, 2019, for all endpoints, except for mortality, where follow-up ended December 31, 2020), whichever came first.

## Statistics

Postoperative weight-loss is presented as change in BMI (BMI-loss = initial BMI – postoperative BMI), TWL (TWL = 100 x weight loss/preoperative weight), and excess BMI-loss (EBMIL = 100 x [initial BMI – postoperative BMI]/[initial BMI – 25]). Categorical data are presented as numbers (n) and percentage (%), continuous variables as mean +/- standard deviation (SD), or median with interquartile range (IQR) as appropriate. The balance between the matched groups was evaluated by calculating the standardized difference. A standardized difference of >0.1 was considered residual imbalance. Binary outcomes were evaluated using logistic regression adjusted for matching variables with Odds ratios (OR) with 95% confidence intervals (95%CI) as measures of association. The occurrence of long-term outcomes was estimated as incidence rates (IR) and further evaluated using Cox-regression adjusted for matching variables with Hazard ratios (HR) and 95%CI as measures of association. Continuous outcomes were evaluated using a linear regression model adjusted for matching variables or Mann-Whitney U-test, as appropriate.

IBM SPSS version 25 (IBM, Armonk, NY,US) and R version 4.0.0 (R Core Team, Vienna, Austria) were used for statistical analyses.

## Ethics

The study was approved by the Swedish Ethical Review Authority (Ref:2022-05359-01). The approval from the Swedish Ethical Review Authority waived informed consent. However, all patients were informed of the registries and that data from these registries can be used for research. They were allowed to withdraw their consent at any time and have the data removed ("opt-out").

## Results

During the study period, 60,114 patients meeting the inclusion criteria were identified. Before surgery, patients with social anxiety disorder were younger, more often women with higher BMI, had a lower income, and were more often of Swedish origin [Supplementary Table S1 in S1 File]. The Propensity score match resulted in two groups without any clinically relevant difference in matched characteristics [Table 1]. Patients with social anxiety disorder more often

**Table 1. Baseline characteristics of patients with social anxiety disorder and matched controls undergoing metabolic and bariatric surgery.**

| | Social anxiety group | Control group | Standardized mean difference[1] |
|---|---|---|---|
| N | 586 | 5791 | |
| Age, years | 36.4 ± 10.2 | 36.2 ± 11.7 | 0.018 |
| Body Mass Index, kg/m2 | 43.2 ± 6.2 | 43.1 ± 5.9 | 0.017 |
| **Sex** | | | |
| Men | 114 (19.5%) | 1066 (18.4%) | 0.028 |
| Women | 472 (80.5%) | 4725 (81.6%) | 0.028 |
| **Comorbidities** | | | |
| Type-2 diabetes | 80 (13.7%) | 726 (12.5%) | 0.036 |
| Hypertension | 93 (15.9%) | 865 (14.9%) | 0.028 |
| Sleep apnea | 51 (8.7%) | 481 (8.3%) | 0.014 |
| Dyslipidemia | 48 (8.2%) | 431 (7.4%) | 0.030 |
| Chronic Obstructive Pulmonary Disorder | 48 (3.8%) | 207 (3.6%) | 0.011 |
| Cardiovascular disease | 10 (1.7%) | 81 (1.4%) | 0.024 |
| **Socioeconomic factors** | | | |
| Disposable income | | | |
| Quartile 1 | 330 (56.3%) | 3173 (54.8%) | 0.030 |
| Quartile 2 | 167 (28.5%) | 1681 (29.0%) | 0.011 |
| Quartile 3 | 60 (10.2%) | 672 (11.6%) | 0.045 |
| Quartile 4 | 29 (4.9%) | 265 (4.6%) | 0.014 |
| Unemployment | 67 (11.4%) | 666 (11.5%) | 0.003 |
| **Origin** | | | |
| Swedish born, Swedish-born parents | 524 (89.4%) | 5175 (89.4%) | 0.002 |
| 2st generation immigrant | 23 (3.9%) | 238 (4.1%) | 0.010 |
| 1st generation immigrant | 39 (6.7%) | 378 (6.5%) | 0.008 |
| **Surgical method** | | | |
| Roux-en-Y gastric bypass | 501 (85.5%) | 4918 (84.9%) | 0.017 |
| Sleeve gastrectomy | 85 (14.5%) | 873 (15.1%) | 0.017 |
| **Surgical access** | | | |
| Laparoscopy | 565 (96.4%) | 5587 (96.5%) | 0.005 |
| Conversion to open surgery | 4 (0.7%) | 41 (0.7%) | 0.003 |
| Primary open surgery | 17 (2.9%) | 163 (2.8%) | 0.006 |

1-Standardized mean difference >0.10 was considered to represent residual imbalance.

received treatment with selective serotonin reuptake inhibitors or serotonin and norepinephrine reuptake inhibitors (n = 408, 69.6% vs. n = 1612, 27.8%) and benzodiazepine use (n = 157, 26.8% vs. n = 348, 6.0%) in the year before surgery. Previous episode of alcohol or substance use disorder was also more common among patients with social anxiety disorder (n = 111, 18.9% vs. n = 376, 6.5%).

## Follow up

Follow-up attendance at 30 days, 1, and 2 years after surgery was similar between the two groups (30 days 97.4% in the social anxiety group vs 97.7% in the control group, p = 0.484; 1 year 84.0% vs 86.0%, p = 0.107; 2 years 59.7% vs 63.6%, p = 0.068). Mean follow-up time for mortality, MACE, substance use, and self-harm was 6.69 ± 2.67 years for patients with social anxiety disorder vs 6.70 ± 2.61 years for controls.

## Weight

Patients with social anxiety disorder had a slightly lower weight-loss at 1 year after surgery, but there were no differences in total weight loss, excess BMI loss or BMI loss at 2 years after surgery [Table 2].

## Early postoperative complications

Patients with social anxiety disorder had a higher risk for postoperative complications within 30 days compared to the control group (OR 1.59; 95%CI 1.21–2.09). An increased risk was mainly seen for abdominal pain (OR 3.00; 95%CI 1.63–5.51) and wound complications (OR 2.19; 95%CI 1.25–3.83). No statistically significant difference was seen for serious postoperative complications (OR 1.39; 95%CI 0.89–2.17) [Supplementary Table S2 in S1 File].

## Mortality and major adverse cardiovascular events

During the study, there were 14 deaths among patients with social anxiety disorder (IR 3.6, 95%CI 2.0–6.0; HR 1.12; 95 CI 0.64–1.95 p=0.685) and 131 in the control group (IR 3.4, 95%CI 2.8–4.0). MACE occurred for 8 patients in the social anxiety group (IR 2.3, 95%CI 1.0–4.6; HR 1.81, 95%CI 0.85–3.83; p=0.123) and for 62 patients in the control group (IR 1.7, 95%CI 1.3–2.1).

## Substance use and self-harm

Alcohol or substance abuse was diagnosed among 111 patients with social anxiety disorder (IR 32.1/1000 person-years; 95%CI 26.4–38.6) and 483 patients in the control group (IR 13.1/1000 person-years; 95%CI 12.0–14.4) corresponding to a more than twofold increased risk of alcohol or substance abuse (HR 2.41, 95%CI 1.96–2.96, p<0.001). New onset of alcohol or substance abuse among patients with no such diagnosis before surgery were diagnosed for 45 patients with social anxiety disorder (IR 15.1/1000 person-years; 95%CI 11.0–20.3) compared to 127 patients in the control group (IR 3.6/1000 person-years; 95%CI 3.0–4.3) corresponding to a twofold increased risk of alcohol or substance abuse (HR 2.19, 95%CI 1.69–2.86, p<0.001).

A self-harm event occurred for 59 patients with social anxiety disorder (IR 16.0/1000 person-years, 95%CI 12.2–20.6) and 246 instances in the control group (IR 6.5/1000 person-years, 95%CI 5.7–7.4) corresponding to a more than twofold increased rate of self harm (HR 2.44 CI 95% 1.84–3.25 p<0.001).

**Table 2. Weight loss at 1 and 2 years after surgery among patients with social anxiety disorder and a matched control group.**

|  | Social anxiety group | Control group | P* |
|---|---|---|---|
| Weight loss at 1 year after surgery |  |  |  |
| BMI-loss, kg/m$^2$ | 13.6±4.23 | 14.0±4.48 | 0.010 |
| Excess BMI loss (%) | 79.2±27.0 | 81.0±24.1 | 0.088 |
| Total weight loss (%) | 31.5±9.1 | 32.3±8.2 | 0.019 |
| Weight loss at 2 years after surgery |  |  |  |
| BMI-loss, kg/m$^2$ | 14.3±5.19 | 14.1±4.94 | 0.974 |
| Excess BMI loss (%) | 82.1±28.4 | 81.3±25.7 | 0.460 |
| Total weight loss (%) | 32.8±10.3 | 32.6±9.7 | 0.911 |

*- Adjusted for matching variables.

## Health-related quality of life

Patients with social anxiety disorder reported worse health-related quality of life at baseline and over 1 and 2 years after surgery [Fig 1].

## Subgroups analyses

Further subgroup analyses were conducted to evaluate the results, stratified by surgical method. The results of these subgroup analyses did not differ from those presented for the merged group (Tables S3-S8 in S1 File).

## Discussion

In this matched cohort study, patients diagnosed with social anxiety disorder had comparable weight loss and follow-up attendance during 2-year follow-up, but an increased risk for non-serious early postoperative complications, as well as alcohol or substance abuse and self-harm over a mean follow-up time of 6.7 years after RYGB and SG, compared to matched controls.

Previous studies have reported an association between anxiety disorders and a sedentary lifestyle [23,24]. A sedentary lifestyle may influence hunger and eating regulation after MBS [25]. Still, no difference in average maximum weight-loss was seen between patients with social anxiety disorder and matched controls in the current study, which supports the reports of two smaller previous studies, both using preoperative anxiety scales to define anxiety disorders [26,27]. While the definition of social anxiety disorder in the present study included a diagnosis of social anxiety disorder in specialized health care suggesting a moderate to severe disease in contrast to a more mild disease in the studies by Hsu and

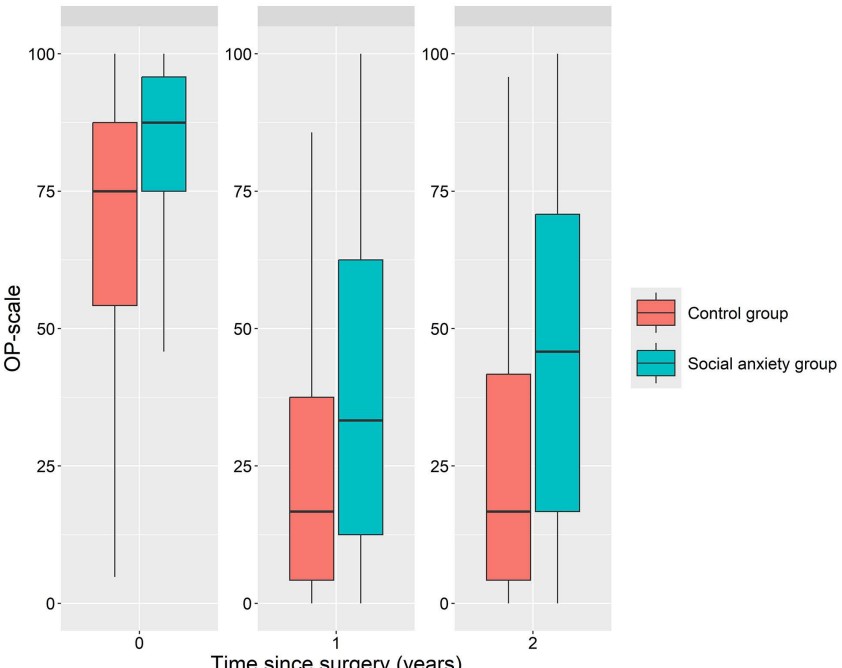

**Fig 1. Health-related quality of life estimated with the Obesity-related problem scale before and after surgery among patients with social anxiety disorder and a matched control group.** The Obesity-related problem scale assesses the impact of obesity on psychosocial functioning on a scale from 0 to 100. Higher scores indicate more psychosocial dysfunction. Median preoperative score in the social anxiety group was 87.5 (Interquartile range 75.0–95.8), and 75.0 (54.2–87.5) in the control group. The levels improved at 1 and 2 years after surgery but remained higher among patients with social anxiety disorder [1 year 33.3 (12.5–62.5) vs. 16.7 (4.2–37.5); 2 years 45.8 (16.7–70.8) vs. 16.7 (0.0–41.7), p<0.001 for all].

Sockalingham, the results of all three studies suggest good weight-loss results during the first years after surgery among patients with social anxiety disorder of varying severity [17,26,27].

A higher overall complication rate was observed in patients with social anxiety disorder. The main difference was noted in non-serious complications, particularly abdominal pain and wound complications. Wound complications encompassed both wound infections and wound-related bleeding. While serotonin reuptake inhibitors may modulate local inflammation, potentially benefiting wound healing, they also inhibit clot formation, thereby increasing the risk of postoperative bleeding, especially when combined with concomitant anticoagulant therapy [28–31]. Higher levels of anxiety have also been associated with increased postoperative impact of somatic symptoms, including pain after RYGB [32]. Behavioural factors, including health-seeking behaviours, may also contribute to the higher incidence of non-serious complications.

Anxiety disorders, including social anxiety, may present complex patterns of both health avoidance and health-seeking behaviour. In the postoperative context, a combination of seeking reassurance and heightened somatic symptoms related to anxiety may tilt the balance toward increased health-seeking behaviour [33]. Furthermore, the perception of postoperative pain is significantly influenced by psychological factors such as anxiety, catastrophizing, and negative expectations [34]. Preoperative anxiety has been previously linked to increased postoperative pain following various surgical procedures, including MBS [27,35,36]. Postoperative interventions with a focus on psychological support after surgery for patients with negative cognition and anxiety can reduce subacute postoperative pain, and thereby be beneficial for selected patients [37].

Consistent with findings for patients with ADHD, social anxiety disorder was associated with an elevated risk of substance abuse and self-harm [11]. The increased incidence of depression, self-harm, and alcohol abuse following MBS is well-documented as some of the more problematic long-term side effects of the procedures [10], and was observed after both RYGB and SG. While an increased health-seeking behavior and established contacts with specialized psychiatric care may contribute to higher diagnostic rates of these conditions among patients with social anxiety disorder, enhanced awareness and support from social networks and healthcare providers outside of specialized psychiatric settings are crucial in identifying mood disorders and early signs of alcohol overconsumption. However, the social avoidance characteristics of social anxiety disorder may complicate the provision of such support. Particular attention from healthcare providers to ensure adequate follow-up and support for these patients may therefore be necessary after MBS. During both preoperative and postoperative contacts, it is important to recognize that obesity as well as social anxiety disorder are each associated with social stigma [38]. This stigma can exacerbate symptoms, including negative self-image and social fears, leading to social isolation and avoidant behaviors. Therefore, follow-up care for patients with social anxiety disorder may require a multidisciplinary approach that promotes health-supportive behaviors, addresses environmental support, continuously challenges misconceptions, and provides non-judgmental monitoring of adverse outcomes such as substance use and risk factors for self-harm.

Patients with social anxiety disorder reported lower health-related quality of life (HRQoL) both preoperatively and at 1- and 2-year follow-up assessments. HRQoL was evaluated using the Obesity-related Problems Scale, which measures the psychosocial impact of obesity. Improvements on this scale correlate strongly with postoperative weight loss [22]. However, since weight loss was comparable between the groups in this study, the observed differences in HRQoL are unlikely to be attributed solely to weight loss. Previous studies have shown that anxiety disorders are associated with less favorable HRQoL trajectories, particularly in the mental health domains [39]. The Obesity-related Problems scale focuses on aspects of psychosocial functioning, an area problematic to individuals with social anxiety. Despite consistently more problematic scores across all time points, it is important to highlight that a significant improvement was observed during the first year post-surgery. Although a higher degree of regression toward preoperative scores was noted at the 2-year follow-up for patients with social anxiety disorder compared to the control group, these patients still exhibited significant improvements, indicating symptom reduction related to social anxiety. Further qualitative and quantitative research, including evaluation of disease severity and other supportive interventions beyond pharmacological treatment, is needed to explore the direct effects of MBS on social anxiety symptoms in more depth.

The current study did not address the mechanism of the connection between social anxiety disorder and differences in outcome after MBS. Further, given the strong link between social anxiety disorder and other psychiatric comorbidities, these were not included in the matching. While social anxiety disorder increases the risk for both alcohol use disorder and self-harm/suicide attempts, [40,41] it is possible that the high burden of psychiatric comorbidities in this cohort overestimated the risk for these adverse outcomes.

The major strength of this study lies in the large nationwide cohort of patients with data from several high-quality registries and the closely matched control group. There are, however, several limitations that must be considered. First, social anxiety disorder was defined by a registered diagnosis in specialised healthcare, which has been reported to include social anxiety disorder of moderate to high severity [17]. In real life, social anxiety disorder exists on a spectrum of severity. No information from symptom severity scales was available. Patients with less significant symptoms are likely to be missed and could therefore result in a slight underestimation of differences between the two groups. However, with an estimated prevalence not exceeding 18% of social anxiety disorder, it is unlikely that this would result in missed differences of clinical relevance. Further, the study mainly includes a population of white Caucasian origin living in Scandinavia. The results may not necessarily be transferable to other ethnic groups and other societies. The study also included patients with social anxiety disorder who agreed to undergo MBS, and by definition had a stable psychiatric situation (with a majority receiving active pharmacological treatment for anxiety). This limits the generalizability to patients with controlled, moderate to severe anxiety disorder who ultimately agree to undergo this type of surgery. As with all real-world studies, it is difficult to retain high follow-up rates over time. Due to the high levels of missing data over long-term follow-up, it was not possible to evaluate weight regain. While similar mid-term weight outcomes are reassuring, they do not guarantee long-term weight loss results. Weight outcomes over long-term follow-up remain an important outcome that should preferably be evaluated in future studies. Finally, despite achieving a close match between cases and controls, with the non-randomized design, the study can only evaluate associations and not causation.

## Conclusion

MBS appears to be a safe and effective treatment for severe obesity in patients with social anxiety disorder. An observed increase in minor postoperative complications, as well as a higher risk for self-harm, and alcohol or substance abuse, should not deter clinicians from considering MBS in this population. An individualized and increased peri- and postoperative support should be considered for patients with moderate to severe social anxiety disorder and severe obesity.

## Supporting information

**S1 File. Table S1**. Baseline characteristics of patients with social anxiety disorder and matched controls undergoing metabolic and bariatric surgery. **Table S2.** Postoperative complications within 30 days after surgery among patients with social anxiety disorder and a matched control group. **Table S3**. Baseline characteristics of patients with social anxiety disorder and matched controls undergoing metabolic and bariatric surgery, stratified by surgical method. **Table S4.** Weight loss at 1 and 2 years after Roux-en-Y gastric bypass surgery among patients with social anxiety disorder and a matched control group. **Table S5.** Weight loss at 1 and 2 years after sleeve gastrectomy among patients with social anxiety disorder and a matched control group. **Table S6.** Risk for alcohol or substance abuse, and self-harm events after surgery among patients with social anxiety disorder compared to a matched control group, stratified by surgical method. **Table S7.** Health-related quality of life estimated with the Obesity related problem scale before and after Roux-en-Y gastric bypass surgery among patients with social anxiety disorder and a matched control group. **Table S8.** Health-related quality of life estimated with the Obesity related problem scale before and after Sleeve gastrectomy among patients with social anxiety disorder and a matched control group.
(DOCX)

## Author contributions

**Conceptualization:** Erik Näslund, Henrik Larsson, Erik Stenberg.

**Data curation:** Erik Stenberg.

**Formal analysis:** Jesper Christiansen, Erik Stenberg.

**Funding acquisition:** Erik Näslund, Erik Stenberg.

**Investigation:** Jesper Christiansen, Erik Stenberg.

**Methodology:** Henrik Larsson, Erik Stenberg.

**Project administration:** Erik Stenberg.

**Resources:** Erik Näslund, Erik Stenberg.

**Software:** Jesper Christiansen.

**Supervision:** Erik Stenberg.

**Visualization:** Erik Stenberg.

**Writing – original draft:** Jesper Christiansen, Erik Stenberg.

**Writing – review & editing:** Erik Näslund, Henrik Larsson.

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
