## [Decision Letter · Decision Letter 0]

13 Jul 2025

Dear Dr. Stenberg,

Thank you for submitting your manuscript to PLOS ONE. After careful consideration, we feel that it has merit but does not fully meet PLOS ONE’s publication criteria as it currently stands. Therefore, we invite you to submit a revised version of the manuscript that addresses the points raised during the review process.

I personally believe this is an important topic, and your study addresses it with originality and methodological rigor. Nevertheless, several issues still need to be resolved. All the reviewers are seasoned clinicians and experienced academicians; as such, their opinions are highly regarded and their insights are valued. I have no further comments. Please address all reviewers' feedback in a point-by-point manner and resubmit your manuscript.

We look forward to receiving your revised manuscript.

Kind regards,

Athanasios G. Pantelis

Academic Editor

PLOS ONE

Journal Requirements:

5. You have indicated that data is available from [soreg@regionorebrolan.se, Registerservice@socialstyrelsen.se , microdata@scb.se]. Please can we ask you to provide us with a general contact email address for the data requests, so readers can request access in perpetuity. If a general email is not available please provide a link to a website where readers can obtain access to data.

6. We note that the grant information you provided in the ‘Funding Information’ and ‘Financial Disclosure’ sections do not match.

7. Thank you for stating the following financial disclosure:

this work was supported by grants from Region Örebro County, Åke Wiberg Foundation, Stockholm County Council, and SRP Diabetes

8. Please remove all personal information, ensure that the data shared are in accordance with participant consent, and re-upload a fully anonymized data set.

Reviewers' comments:

Reviewer's Responses to Questions

**Comments to the Author**

1. Is the manuscript technically sound, and do the data support the conclusions?

Reviewer #1: Yes

Reviewer #2: Yes

Reviewer #3: Yes

Reviewer #4: Yes

Reviewer #5: Yes

2. Has the statistical analysis been performed appropriately and rigorously?

Reviewer #1: No

Reviewer #2: Yes

Reviewer #3: Yes

Reviewer #4: Yes

Reviewer #5: Yes

3. Have the authors made all data underlying the findings in their manuscript fully available?

Reviewer #1: Yes

Reviewer #2: Yes

Reviewer #3: Yes

Reviewer #4: Yes

Reviewer #5: No

4. Is the manuscript presented in an intelligible fashion and written in standard English?

Reviewer #1: Yes

Reviewer #2: Yes

Reviewer #3: Yes

Reviewer #4: Yes

Reviewer #5: Yes

Reviewer #1: Major Concerns

• While the propensity matching was comprehensive for many demographic and medical factors, the SAD group differed markedly from controls in terms of baseline psychiatric profile. Notably, 69.6% of patients with SAD were on an SSRI/SNRI antidepressant (versus 27.8% of controls) and 26.8% were using benzodiazepines (versus 6.0% of controls) in the year before surgery . Prior diagnoses of substance use disorder were also more common in the SAD cohort (18.9% vs 6.5% in controls) . These differences reflect a higher burden of underlying psychiatric illness (e.g. depression, generalized anxiety) in the SAD group. Such comorbidities and treatments could themselves influence outcomes like self-harm and substance abuse relapse, independent of SAD per se. The authors have identified these baseline differences in their Results , but there is concern that they did not fully adjust for or discuss their impact. The analytic approach adjusted for the matching variables, yet psychiatric medication use and prior psychiatric history were not matching factors and thus remained imbalanced . This raises the possibility that the observed excess of self-harm and substance-related events in the SAD group might be partly driven by these pre-existing comorbid conditions rather than social anxiety alone. To strengthen their conclusions, the authors should consider additional analyses or discussion to account for these factors. For example, a sensitivity analysis controlling for baseline antidepressant use, benzodiazepine use, or prior substance abuse history would help determine if SAD remains an independent predictor of postoperative self-harm/substance outcomes after accounting for these covariates. At minimum, a more explicit discussion acknowledging that comorbid psychiatric conditions (and their treatments) could confound the relationship between SAD and adverse outcomes would be valuable.

• The study defines SAD exposure via a recorded diagnosis in specialized healthcare, which by design captured moderate-to-severe cases and likely missed milder cases . This criterion ensures the cohort truly had clinically significant social anxiety, but it may introduce a form of selection bias. Patients with mild social anxiety (or those never diagnosed) were included among “controls,” potentially diluting group differences. More importantly, patients with the most disabling social anxiety might be underrepresented in the surgical cohort altogether – for instance, some individuals with severe SAD might avoid frequent medical appointments or opt out of surgery due to anxiety about the clinical process. The authors did not directly address how many surgery candidates with psychiatric issues drop out or are denied clearance for psychological reasons. Previous research has noted that psychological factors (including anxiety) can lead a subset of bariatric candidates to drop out before surgery . Thus, the studied population of SAD patients likely represents those who were able and willing to undergo surgery, which could mean they had adequate support or slightly less impairment in functioning than the “average” person with severe SAD. This potential selection limits generalizability – the outcomes observed may not fully apply to all individuals with SAD and obesity, especially those who never made it to surgery. The authors should discuss this context. For example, they might note that their findings apply to patients with SAD who successfully complete the surgical evaluation process, and that unmeasured factors (like level of social support or motivation) might differentiate those who underwent surgery from those who did not. A related issue is the study’s demographic specificity: the cohort was almost entirely ethnically homogenous (predominantly Caucasian in Sweden) . The authors do acknowledge this and the limited transferability to other ethnic and cultural contexts . We agree with that self-acknowledged limitation and further encourage clarifying that healthcare system differences (e.g. the structured follow-up in Sweden) might mean results could differ elsewhere.

• The study’s mean follow-up of ~6.7 years for certain outcomes is a strength, yet complete longitudinal data on weight and HRQoL were only available up to 2 years post-surgery. The authors state that high missing data in long-term follow-up prevented evaluation of weight regain . This is a major limitation because weight regain after the first 1–2 years is common and could differ between groups. If patients with anxiety have more difficulty sustaining behavioral changes long-term, one might expect differential weight maintenance beyond the 2-year mark. In fact, external evidence suggests anxiety symptoms can predict weight outcomes over time – for example, one study found that patients with high preoperative anxiety initially lost as much weight as others at 1 year, but then regained more weight by 30 months, leading to less total weight loss at 2.5 years post-op . The present study’s finding of equivalent 2-year weight loss in SAD vs control groups is reassuring, but without data on later follow-ups we cannot know if that parity persists. The inability to assess weight regain means the long-term efficacy of MBS in SAD patients (in terms of sustained weight control) remains unaddressed. This should be highlighted as a significant limitation. The authors might consider tempering their conclusion that outcomes are “similar” in the long run, since it’s based on relatively short-term weight results. Encouragingly, they do emphasize the need for future studies on weight regain . We concur and suggest that incorporating longer-term monitoring (perhaps leveraging registry updates or linking to other data sources) would greatly enhance understanding of how anxiety disorders might impact durability of weight loss. If such data cannot be obtained for this cohort, the authors should still acknowledge that two-year outcomes do not guarantee long-term outcomes, especially in light of literature hinting at possible divergences beyond that timeframe.

• The finding of increased “non-serious” postoperative complications in the SAD group is interesting, but some clarification is needed on this outcome. The manuscript defines a serious complication rigorously (using Clavien-Dindo ≥ IIIb: requiring reoperation, organ failure, or death) , and by inference “non-serious” complications are those below that threshold. It appears that minor complications – particularly wound issues (infections or bleeding) and abdominal pain – drove the higher complication rate in SAD patients . The authors offer a plausible hypothesis that SSRIs (taken by a large proportion of SAD patients) might contribute to bleeding risk or impaired clotting, especially if combined with anticoagulants. This mechanistic speculation is valuable. However, it would be useful to report or discuss whether differences in minor complications could also stem from behavioral factors. For instance, could patients with anxiety have a lower threshold for seeking care for issues like pain, or might they be more closely monitored, leading to higher detection of minor problems? The manuscript does not currently address this aspect. Additionally, we should ensure that the analysis of “serious” complications was not biased by missing data: since serious complications were only tracked starting in 2010 , patients who had surgery in 2007–2009 (a subset of the cohort) might not have complete data on that outcome. If those early cases were included, some serious events could be unreported, potentially underestimating serious complication rates. It’s not clear if the authors excluded 2007–2009 cases from the serious complication analysis or imputed those data. Clarifying this in the Methods would be helpful for transparency. Overall, the complication findings are noteworthy but would benefit from a bit more context. I suggest the authors explicitly note in the Discussion that no increase in life-threatening complications was observed (which is an important reassuring point), and elaborate on why minor complications were higher. Discussing both biological factors (e.g. medication effects on wound healing ) and possible psychosocial factors (e.g. health-seeking behavior) would provide a more rounded interpretation.

Minor Concerns

• The manuscript is generally well-written, but a few sentences need minor editing for grammar/typos. For example, in the Abstract the line “patients with social anxiety disorder significantly improvement in health-related quality of life” is grammatically incorrect – it should read “experienced a significant improvement”. In the Discussion, one sentence reads “…diagnosis in specialized health care sugging a moderate to severe disease…” , where presumably “sugging” is a typo for “suggesting.” These small errors can distract readers and should be fixed. A careful proofread to catch minor spelling/grammatical issues (such as missing articles or pluralization issues) would improve overall readability. Aside from typos, most of the text is clear; just ensure consistency in verb tenses and that each abbreviation is defined at first use (HRQoL was defined properly , as was MBS, etc.).

•Data Presentation – P-values and Significance: In Table 1 (baseline characteristics), some variables show very small differences between groups (e.g. a 0.2% difference in the proportion with hypertension or a 0.1 kg/m² difference in BMI) accompanied by p-values < 0.05 . The authors correctly note that there were no clinically relevant differences post-matching . It may help to explicitly indicate in a footnote that due to the large sample size, trivial differences can become statistically significant. Emphasizing clinical significance over p-value could prevent readers from overinterpreting those p-values. Similarly, when reporting outcomes, consider stating absolute rates in addition to hazard/odds ratios. For instance, the text could mention the actual incidence of self-harm in each group (e.g. “3.6 vs 1.7 events per 1000 person-years” from the results ) alongside the HR 2.44, to give the reader a concrete sense of risk difference. Minor additions like this would enhance the presentation.

• The manuscript alludes to follow-up adherence as an important factor and notes a slight trend toward lower follow-up attendance at 2 years for SAD patients (59.7% vs 63.6%, p=0.068) . It would be worth reporting the 1-year and 2-year follow-up rates more explicitly, perhaps in the Results or a table, since follow-up compliance is a key aspect of postoperative care. Currently these numbers appear in passing. Clarifying that there was no significant difference at 1 year (both ~86% follow-up) and only a non-significant drop-off difference at 2 years is useful information; it suggests that despite their social anxiety, these patients managed to engage with early follow-up nearly as well as controls. This detail supports the discussion point that careful follow-up is feasible and important in SAD patients, so highlighting it would be beneficial.

•Ensure consistent use of terms like “MBS” (metabolic and bariatric surgery). In some places the manuscript simply says “bariatric surgery” – which is fine – but since the title and much of the text use MBS, stick to one convention or clearly state that MBS encompasses metabolic surgery as well. This is a very minor point, but consistency avoids any confusion. Likewise, abbreviations like HRQoL, MACE (major adverse cardiovascular events), etc., are defined in the methods ; just be sure each appears with definition on first use in the main text and perhaps again in tables if they stand alone.

•A minor formatting observation – the reference list includes a StatPearls chapter (ref. 3) and some multi-line entries. Just ensure that all references follow journal style. For example, Reference 4 (Rosik, 2005) appears to be broken into two lines in the PDF ; when finalizing, check that each reference is properly concatenated. This is likely just a artifact of the PDF formatting, but worth a quick look to make sure no citations are inadvertently duplicated or truncated.

Limitations Not Acknowledged by Authors

While the authors have an appropriate limitations section discussing several key issues, there are a few additional limitations that were not explicitly addressed:

• Potential Misclassification of Controls: The study assumes that the control group has no social anxiety disorder, but because the registry only captures diagnosed cases in specialist care, some control patients might actually have undiagnosed or mild social anxiety. Such misclassification would bias the comparison toward the null (making groups more similar than assumed). The authors touched on the possibility of missed mild cases leading to underestimation of differences . They could make it more explicit that a proportion of “controls” could have subclinical anxiety that went unrecorded. This limitation is somewhat inherent to using ICD diagnoses – it might be worth acknowledging that SAD was treated as a binary, present/absent variable, whereas in reality anxiety symptoms exist on a spectrum. If some controls had social anxiety symptoms (just not formally diagnosed or treated), the true impact of having significant SAD could be larger than observed. A brief comment to this effect would be appropriate.

• Influence of Treatment for SAD: As noted earlier, many patients in the SAD cohort were receiving pharmacotherapy (antidepressants, anxiolytics). The therapeutic management of their anxiety could confound outcomes in different directions – for example, SSRIs might reduce some anxiety-related complications but contribute to other issues (like the wound healing discussion). The study does not separate the effects of the disorder from the effects of its treatment. One could imagine that patients with adequately treated anxiety might fare differently than those with untreated anxiety. The data to directly analyze this may not be available, but the authors should acknowledge that treatment status was not controlled. For instance, is it possible that the subset of SAD patients not on SSRIs (30% of them) had worse outcomes than those on medication? Or vice versa? This is speculative, but since virtually all SAD patients were managed for their condition in some way, the reported outcomes reflect the combination of having SAD and receiving standard psychiatric care for it. The authors might clarify that their findings apply to patients with SAD under treatment, and that outcomes could differ if a patient’s SAD were unrecognized or untreated. Recognizing this nuance is important for external validity – it suggests that ensuring patients are in treatment for their anxiety (as they were in Sweden) might be one factor in achieving good weight loss results.

• Lack of Direct Anxiety Outcome Measures: The study infers improvement in social anxiety symptoms from the HRQoL and the context, but it did not measure social phobia severity or anxiety levels directly after surgery. There was no specific psychiatric assessment (e.g. a social anxiety scale) administered postoperatively. This is understandable given the registry nature, but it means we don’t know how social anxiety per se changed, aside from the proxy of improved psychosocial quality-of-life. The authors themselves call for qualitative research to explore direct effects on SAD symptoms . This could be mentioned as a limitation: the study cannot confirm if bariatric surgery led to a reduction in social anxiety symptomatology, since that wasn’t directly measured. The observed HRQoL gains suggest some relief in social functioning, but targeted studies would be needed to conclude that SAD severity decreases post-surgery. By acknowledging this, the authors underline that their claim of surgery being “safe and effective” in SAD is chiefly about physical and general quality-of-life outcomes, not a cure of the anxiety disorder itself.

• Unmeasured Psychosocial Variables: Beyond the presence of SAD and other diagnoses, there could be other psychosocial factors not captured in the datasets that influence outcomes. For example, level of social support, coping skills, or therapist involvement post-surgery were not recorded. If SAD patients differed systematically in these (perhaps lower support networks due to their social fears, as the discussion suggests ), that could affect follow-up and mental health outcomes. The authors hint at this by discussing how social avoidance can complicate support provision . It would strengthen the limitations section to note that important factors like adherence to counseling, support group participation, or family support were not available in the registries. These could moderate outcomes (e.g. a well-supported SAD patient might not experience self-harm, whereas an isolated one might).

• Subgroup Analyses: The cohort included both primary surgery types (gastric bypass and sleeve gastrectomy) and both sexes, etc. The study did not report any subgroup analyses (e.g. did the effect of SAD on outcomes differ for gastric bypass vs sleeve, or for younger vs older patients?). While not strictly necessary, the absence of subgroup analysis could be noted as a limitation if interactions might be expected. For instance, alcohol misuse risk is known to be higher after gastric bypass than sleeve in general. If a large portion of the SAD group had bypass, could that partly explain the high substance use incidence? The matching was stratified by procedure, so presumably distribution was equal, and the overall HR for substance abuse was averaged across procedures. If the data allow, exploring whether one surgery type is riskier for SAD patients (in terms of mental health outcomes) could be insightful. If not, it’s a missed opportunity but not a flaw – however, mentioning that the study wasn’t powered to examine differences by surgery subtype or other subgroups would be a fair caveat.

Language and Style

There are only a few instances where language could be polished:

• As noted under minor concerns, there are a handful of typos/grammatical errors (e.g. “significantly improvement” instead of “significant improvement,” “sugging” instead of “suggesting”) . These appear to be simple mistakes that slipped through proofreading. Correcting these will enhance professionalism.

• A bit of attention to concise phrasing could help in a few spots. For example, the sentence in Results about SSRIs could be rephrased for clarity: “the propensity match resulted in two groups without clinically relevant differences in matched variables, but patients with SAD had higher use of SSRIs/SNRIs (69.6% vs 27.8%) and benzodiazepines (26.8% vs 6.0%) in the year before surgery” . Breaking it into two sentences or using parentheses for the percentages might improve readability. This is a minor stylistic suggestion.

• The Discussion is comprehensive, but a couple of sentences are quite long. For instance, the sentence spanning lines 923–931 has multiple clauses (“Enhanced awareness and support from social networks and healthcare providers are crucial… however, the social avoidance… necessitating particular attention…”). It’s an important point; splitting it into two sentences would make it easier to follow. In general, ensure each sentence conveys one clear idea.

• Consistency in tone: The tone is appropriately academic. In a few places, the wording could be slightly more precise. For example, “MBS appears to be a safe and effective treatment for severe obesity in patients with social anxiety disorders” – since “safe and effective” is a strong claim, the authors might consider adding “in the short-to-medium term” or similar, acknowledging the time frame of their data. But this is more about nuance than language correctness.

Thank you

Reviewer #2: Thank you for the opportunity to review this manuscript. The topic is highly relevant, given the worldwide prevalence of social anxiety. The study is well-structured and its objectives are clearly articulated. I have only one minor language concern: in the Abstract (lines 46-48) the main verb of the sentence is missing and should be supplied.

Furthermore, the paper would be strengthened by including a comparator group of patients with social anxiety who did not undergo metabolic/bariatric surgery (MBS). Such a comparison would highlight more convincingly the potential impact of MBS on this population. The marked improvement in health-related quality of life (HRQL) reported here is striking—especially in a condition where HRQL typically deteriorates over time—and deserves to be contrasted with outcomes in patients who receive no obesity treatment.

In my opinion the conclusion should be re-phrased in order to reflect clearly the intersting findings of this paper.

I would say: "MBS appears to be a safe and effective treatment for severe obesity in patients with social-anxiety disorder. The observed increase in minor postoperative complications—as well as the higher propensity for self-harm and alcohol or substance misuse—should not deter clinicians from recommending MBS to this population; instead, these risks highlight the importance of comprehensive pre-operative screening and structured postoperative follow-up. An individualized..."

In summary, this is a well-conceived and clinically relevant study that addresses theimportant intersection between mental health and metabolic surgery. With minor language corrections and the suggested considerations for future comparison groups, the manuscript would offer an even more compelling contribution to the literature. I congratulate the authors for their work and hope these comments will help strengthen the final version.

Reviewer #3: Dear Authors,

I read your manuscript carefully. I found it well-written and informative. I have some comments and suggestions and I hope addressing them can increase the quality of your upcoming manuscript:

1- Please use "Metabolic and bariatric surgery (MBS)" instead of bariatric surgery in the whole manuscript including Keywords.

2- It can be excellent if you perform a subgroup analysis to compare SG and RYGB on anxiety disorders, HRQoL, and self-harm /suicide/ alcohol or substance abuse after MBS.

3- Some studies reported worsening of psychological disorders and substance abuse after RYGB compared to SG. Please report your results after subgroup analysis and add this point to the discussion part if this finding was found in your study or not.

Reviewer #4: This is a well-written manuscript presenting interesting and unique data. The findings have the potential to improve clinical care for a vulnerable patient group.

I have a few suggestions that I believe would further strengthen the manuscript:

In the paragraph on lines 80–86, please consider adding one or two additional sentences describing the characteristics of social anxiety disorder. Not all readers may be familiar with the condition, and a brief explanation—such as how social anxiety often leads to avoidance behaviors—would be appropriate.

The reasoning in lines 90–92 is somewhat difficult to follow. Please clarify what you mean by the connection to neuropsychiatry.

Regarding the Obesity-related Problems Scale: Please clarify that higher scores indicate greater problems. Most QoL scales are scored so that higher values reflect better quality of life. On line 309, you state that the group with social anxiety had lower scores, but Table 3 shows that they had higher scores, which aligns with the scale’s scoring. I suggest converting Table 3 into a figure, as this would make the results more illustrative. Also, please indicate in the figure legend (or table legend, if you choose not to convert it) that higher scores reflect poorer QoL. Be consistent in how you refer to the Obesity-related Problems Scale—sometimes it is written with a hyphen, sometimes without.

On line 308, when discussing what the Obesity-related Problems Scale measures, I suggest adding a clarification that the scale captures avoidance of social situations due to weight or body shape.

I recommend that you mention, at least briefly in the discussion, the social stigma associated with obesity and how this may interact with and potentially reinforce social anxiety.

Individuals with a psychiatric diagnosis may be expected to have more frequent contact with healthcare services than undiagnosed individuals. Could part of the higher prevalence of self-harm and substance use diagnoses in this group be explained by more frequent follow-up and thus a greater likelihood of receiving a diagnosis?

Please be more specific in your suggestions regarding how this patient group should be followed in clinical care to ensure improved management.

Additional details:

Please spell out what MSD stands for in the conflict of interest section.

Some references are written in parentheses, others are not—please ensure consistency.

Line 277: “sugging” – is this a typo? Should it be “suggesting”?

Line 331: Should it be “disorders” or “disorder”?

Reviewer #5: Summary:

This manuscript reports outcomes after bariatric surgery in patients with moderate to severe social anxiety disorder (SAD), using data from Sweden’s national quality registries. The authors use propensity score matching and longitudinal follow-up to assess weight loss, complications, and psychiatric outcomes. The topic is relevant and addresses an evidence gap regarding mental health comorbidity and surgical outcomes. The methods are standard for this type of question, and the data sources are of high quality. However, several clarifications are needed before the manuscript can be considered for publication.

Major Comments:

1. Definition and Severity of Social Anxiety Disorder

The manuscript refers to patients with “moderate to severe” social anxiety disorder, but it remains somewhat unclear how this was operationalized. Please elucidate this and clarify previous reports on how registry and severity associate.

o Is it known whether the F40.1 ICD-10 diagnosis based on inpatient care, specialist outpatient care, or general practitioner records?

o Were diagnostic criteria or symptom severity scales (e.g., LSAS) available?

o Please clarify the criteria for inclusion and whether the diagnosis reflects clinically verified moderate/severe illness or simply any registered SAD code.

2. Psychiatric Comorbidities and Confounding

While the use of SSRIs and benzodiazepines is reported, other common comorbid psychiatric conditions (e.g., major depression, generalized anxiety, PTSD) are not clearly addressed.

o Please clarify whether patients with additional psychiatric diagnoses were excluded or adjusted for.

o A sensitivity analysis excluding patients with severe psychiatric comorbidities or stratifying by psychiatric medication class would strengthen causal interpretation.

Minor Comments:

1. Use of “Moderate to Severe” Terminology

If there is no standardized way the diagnosis was stratified by severity in the registry, consider avoiding this term or use wording that signifies the limitation of registry diagnosis.

2. Discussion Section

I encourage the authors to discuss the broader clinical implications of their findings a little more, especially how the presence of SAD should prompt specific interventions in this patient population. This would help clinicians to implement changes in their own practice and significantly upgrade the clinical relevance.

**Do you want your identity to be public for this peer review?** For information about this choice, including consent withdrawal, please see our Privacy Policy

Reviewer #1: **Yes:** Mohamed HANY

Reviewer #2: **Yes:** Georgia Doulami

Reviewer #3: **Yes:** Mohammad Kermansaravi

Reviewer #4: No

Reviewer #5: No

---

## [Author Response · Author response to Decision Letter 1]

23 Oct 2025

Dear Editor, Prof. Pantelis

We would like to thank the editorial board and reviewers for their comments on our manuscript. We have responded to each of the comments in the text below and made changes to the manuscript, Tables, and Supplements in response. We strongly believe that the comments from the reviewers have strengthened the manuscript. We hope that the revision matches your expectations and that it can be accepted for publication in PLoS One.

On behalf of all authors,

Erik Stenberg MD, PhD

Journal Requirements:

Author's response: The revision has been updated accordingly

Author’s response: Unfortunately, the ORCID ID is linked to the corresponding authors other address (erik.stenberg@regionorebrolan.se) which is probably the reason to why it could not be accepted by the system. The correct ORCID ID is 0000-0001-9189-0093

Author's response: The full name of the Swedish Ethical Review Authority is stated in the manuscript. As accepted by the Swedish Ethical Review Authority, and in line with current praxis for large, registry-based studies in Sweden, informed consent was waived for the current study which has been clearly stated in the updated version of the manuscript as requested (i.e patients are informed of the registries and that research will be conducted on the data in the registries. They can, at any time, require removal of their data from the registries (“opt-out”)).

Author's response: The current Swedish legislation forbids sharing of pseudonymized data in public domains. However, the data is stored at Örebro University, and can be made available upon reasonable request from researchers. Previously, direct contact with the corresponding author for such requests has not been accepted by journals. Therefore, the official contacts to the registries providing the data has been listed. However, there are two reasonable options:

1. The corresponding author, who has direct access to the stored data, can be listed as contact person

2. The SOReg, which also has access to the dataset (and is an official party) can be listed as contact.

In the revised version, the corresponding author has been listed as contact person for such request, but this can be changed to the official SOReg contact address if the Editors prefer.

5. You have indicated that data is available from [soreg@regionorebrolan.se, Registerservice@socialstyrelsen.se, microdata@scb.se]. Please can we ask you to provide us with a general contact email address for the data requests, so readers can request access in perpetuity. If a general email is not available please provide a link to a website where readers can obtain access to data.

Author's response: Please see response to Q4.

6. We note that the grant information you provided in the ‘Funding Information’ and ‘Financial Disclosure’ sections do not match.

Author's response: These grants have been included in the disclosure to match the reports.

7. Thank you for stating the following financial disclosure:

this work was supported by grants from Region Örebro County, Åke Wiberg Foundation, Stockholm County Council.

Author's response: Thank you! This has been included in the revised cover letter and financial disclosure to match the requirements.

8. Please remove all personal information, ensure that the data shared are in accordance with participant consent, and re-upload a fully anonymized data set.

Author's response: No personal information is included

Author's response: The information has been listed as requested.

Author's response: Thank you for this clarification. We have acted accordingly during the revision.

Reviewers' comments:

Reviewer's Responses to Questions

Comments to the Author

Reviewer #1: Major Concerns

• While the propensity matching was comprehensive for many demographic and medical factors, the SAD group differed markedly from controls in terms of baseline psychiatric profile. Notably, 69.6% of patients with SAD were on an SSRI/SNRI antidepressant (versus 27.8% of controls) and 26.8% were using benzodiazepines (versus 6.0% of controls) in the year before surgery . Prior diagnoses of substance use disorder were also more common in the SAD cohort (18.9% vs 6.5% in controls) . These differences reflect a higher burden of underlying psychiatric illness (e.g. depression, generalized anxiety) in the SAD group. Such comorbidities and treatments could themselves influence outcomes like self-harm and substance abuse relapse, independent of SAD per se. The authors have identified these baseline differences in their Results , but there is concern that they did not fully adjust for or discuss their impact. The analytic approach adjusted for the matching variables, yet psychiatric medication use and prior psychiatric history were not matching factors and thus remained imbalanced . This raises the possibility that the observed excess of self-harm and substance-related events in the SAD group might be partly driven by these pre-existing comorbid conditions rather than social anxiety alone. To strengthen their conclusions, the authors should consider additional analyses or discussion to account for these factors. For example, a sensitivity analysis controlling for baseline antidepressant use, benzodiazepine use, or prior substance abuse history would help determine if SAD remains an independent predictor of postoperative self-harm/substance outcomes after accounting for these covariates. At minimum, a more explicit discussion acknowledging that comorbid psychiatric conditions (and their treatments) could confound the relationship between SAD and adverse outcomes would be valuable.

Author's response: This is a fair point. However, social anxiety disorder is strongly linked to many psychiatric comorbidities. Excluding these, adjusting for them, or including them as matching variables, would over adjust for mental illness, and, from our point of view, not answer the question of what outcomes can be expected for patients with social anxiety disorder. For this reason, we are also reluctant to perform further analyses stratified or adjusted for other aspects of mental health. On the other hand, we agree with the reviewer's comments that the confounding nature of other psychiatric conditions warrants further discussion and have included this in the revised manuscript. In addition, the manuscript includes stratified analyses by previous substance use. (lines 344-349)

• The study defines SAD exposure via a recorded diagnosis in specialized healthcare, which by design captured moderate-to-severe cases and likely missed milder cases . This criterion ensures the cohort truly had clinically significant social anxiety, but it may introduce a form of selection bias. Patients with mild social anxiety (or those never diagnosed) were included among “controls,” potentially diluting group differences. More importantly, patients with the most disabling social anxiety might be underrepresented in the surgical cohort altogether – for instance, some individuals with severe SAD might avoid frequent medical appointments or opt out of surgery due to anxiety about the clinical process. The authors did not directly address how many surgery candidates with psychiatric issues drop out or are denied clearance for psychological reasons. Previous research has noted that psychological factors (including anxiety) can lead a subset of bariatric candidates to drop out before surgery . Thus, the studied population of SAD patients likely represents those who were able and willing to undergo surgery, which could mean they had adequate support or slightly less impairment in functioning than the “average” person with severe SAD. This potential selection limits generalizability – the outcomes observed may not fully apply to all individuals with SAD and obesity, especially those who never made it to surgery. The authors should discuss this context. For example, they might note that their findings apply to patients with SAD who successfully complete the surgical evaluation process, and that unmeasured factors (like level of social support or motivation) might differentiate those who underwent surgery from those who did not. A related issue is the study’s demographic specificity: the cohort was almost entirely ethnically homogenous (predominantly Caucasian in Sweden) . The authors do acknowledge this and the limited transferability to other ethnic and cultural contexts . We agree with that self-acknowledged limitation and further encourage clarifying that healthcare system differences (e.g. the structured follow-up in Sweden) might mean results could differ elsewhere.

Author's response: We agree with the limitations in generalizability and have clarified this limitation further in the discussion. (lines 360-364)

• The study’s mean follow-up of ~6.7 years for certain outcomes is a strength, yet complete longitudinal data on weight and HRQoL were only available up to 2 years post-surgery. The authors state that high missing data in long-term follow-up prevented evaluation of weight regain . This is a major limitation because weight regain after the first 1–2 years is common and could differ between groups. If patients with anxiety have more difficulty sustaining behavioral changes long-term, one might expect differential weight maintenance beyond the 2-year mark. In fact, external evidence suggests anxiety symptoms can predict weight outcomes over time – for example, one study found that patients with high preoperative anxiety initially lost as much weight as others at 1 year, but then regained more weight by 30 months, leading to less total weight loss at 2.5 years post-op . The present study’s finding of equivalent 2-year weight loss in SAD vs control groups is reassuring, but without data on later follow-ups we cannot know if that parity persists. The inability to assess weight regain means the long-term efficacy of MBS in SAD patients (in terms of sustained weight control) remains unaddressed. This should be highlighted as a significant limitation. The authors might consider tempering their conclusion that outcomes are “similar” in the long run, since it’s based on relatively short-term weight results. Encouragingly, they do emphasize the need for future studies on weight regain . We concur and suggest that incorporating longer-term monitoring (perhaps leveraging registry updates or linking to other data sources) would greatly enhance understanding of how anxiety disorders might impact durability of weight loss. If such data cannot be obtained for this cohort, the authors should still acknowledge that two-year outcomes do not guarantee long-term outcomes, especially in light of literature hinting at possible divergences beyond that timeframe.

Author's response: Obtaining long-term follow-up data is certainly a key to further knowledge. However, as we have discussed in the manuscript (and as has been acknowledged by others), maintaining long-term follow-up for large populations after bariatric surgery, in particular for patients with mental health-related conditions, remains very difficult. Unfortunately, there are few other reliable sources for weight in this population. We have further emphasized the importance of long-term weight data to evaluate weight regain / recurrent weight gain for this group of patients. (lines 282 and 366-368)

• The finding of increased “non-serious” postoperative complications in the SAD group is interesting, but some clarification is needed on this outcome. The manuscript defines a serious complication rigorously (u

---

## [Decision Letter · Decision Letter 1]

14 Nov 2025

Dear Dr. Stenberg,

We look forward to receiving your revised manuscript.

Kind regards,

Athanasios G. Pantelis

Academic Editor

PLOS ONE

Journal Requirements:

**Additional Editor Comments:**

Please address all reviewers' comments before proceeding to acceptance for publication.

Reviewers' comments:

Reviewer's Responses to Questions

**Comments to the Author**

Reviewer #1: (No Response)

Reviewer #2: All comments have been addressed

Reviewer #3: All comments have been addressed

Reviewer #4: All comments have been addressed

2. Is the manuscript technically sound, and do the data support the conclusions?

Reviewer #1: Yes

Reviewer #2: Yes

Reviewer #3: Yes

Reviewer #4: Yes

3. Has the statistical analysis been performed appropriately and rigorously?

Reviewer #1: No

Reviewer #2: Yes

Reviewer #3: I Don't Know

Reviewer #4: Yes

4. Have the authors made all data underlying the findings in their manuscript fully available?

Reviewer #1: Yes

Reviewer #2: Yes

Reviewer #3: Yes

Reviewer #4: Yes

5. Is the manuscript presented in an intelligible fashion and written in standard English?

Reviewer #1: Yes

Reviewer #2: Yes

Reviewer #3: Yes

Reviewer #4: Yes

Reviewer #1: While the authors were generally responsive to reviewer feedback, three substantive scientific issues remain incompletely resolved:

1. The reviewer requested sensitivity analyses to determine whether social anxiety disorder remains an independent predictor of adverse outcomes after controlling for baseline psychiatric medication use (antidepressants, benzodiazepines) and prior substance abuse history.

Author Response: Declined to perform the analysis, arguing it would constitute over-adjustment for mental illness intrinsic to the SAD population. Added discussion of confounding and included stratified analysis by substance use.

The concern about whether observed adverse outcomes are attributable to SAD itself versus its associated comorbidities and treatments remains partially unresolved. The stratified analysis by substance use provides some insight but does not fully address the medication confounding issue.

2. Weight and quality of life data only available for 2 years post-surgery, preventing assessment of long-term weight regain patterns that may differ between SAD and control groups.

Author Response: Acknowledged as a significant limitation and emphasized the need for future studies with longer follow-up. Noted the difficulty of maintaining long-term follow-up in this population.

Limits the ability to draw conclusions about long-term efficacy of bariatric surgery in the SAD population, particularly given external evidence suggesting potential divergence in weight trajectories beyond 2 years.

3.: The reviewer suggested exploring whether behavioral factors (e.g., anxiety-related differences in care-seeking behavior, lower threshold for reporting symptoms, or differential monitoring intensity) might contribute to the higher rate of non-serious complications in the SAD group.

Author Response: Discussed biological mechanisms (SSRI effects on bleeding/clotting) but provided limited discussion of behavioral factors.

Partially addressed with focus on biological rather than behavioral mechanisms.

The mechanistic understanding could be enhanced by considering both biological and behavioral pathways, but this does not fundamentally alter the study’s findings or conclusions.

Reviewer #2: Dear Authors,

In my opinion, you have addressed all the comments appropriately. I am satisfied with the manuscript in its current form, acknowledging both its strengths and limitations. I believe the insights it provides are valuable to the metabolic and bariatric surgery community. I look forward to seeing more interesting results from your database in the future.

Kind regards

Reviewer #3: Dear Authors,

Thanks for addressing my comments. I think your manuscript can be publishable in PLOS One journal.

Reviewer #4: All my comments have been addressed.

However, I noticed an inconsistent use of the abbreviation HRQoL, it is sometimes used and sometimes spelled out.

In the figure legend for Figure 1, p < 0.001 for all is stated twice. It should be sufficient to mention it once.

**Do you want your identity to be public for this peer review?** For information about this choice, including consent withdrawal, please see our Privacy Policy

Reviewer #1: No

Reviewer #2: No

Reviewer #3: **Yes:** Mohammad Kermansaravi

Reviewer #4: No

---

## [Author Response · Author response to Decision Letter 2]

23 Dec 2025

Reviewer #1: While the authors were generally responsive to reviewer feedback, three substantive scientific issues remain incompletely resolved:

1. The reviewer requested sensitivity analyses to determine whether social anxiety disorder remains an independent predictor of adverse outcomes after controlling for baseline psychiatric medication use (antidepressants, benzodiazepines) and prior substance abuse history.

“Response: Declined to perform the analysis, arguing it would constitute over-adjustment for mental illness intrinsic to the SAD population. Added discussion of confounding and included stratified analysis by substance use.”

The concern about whether observed adverse outcomes are attributable to SAD itself versus its associated comorbidities and treatments remains partially unresolved. The stratified analysis by substance use provides some insight but does not fully address the medication confounding issue.

Authors response: We acknowledge this comment from the reviewer. Although we agree with the conclusion that the mechanism explaining the risks associated with SAD remains partially unresolved, this cannot fully be evaluated with the current study design, and this was also not part of the aim and design of the current study. This is a limitation that has been discussed within the manuscript and should be a focus of future studies.

2. Weight and quality of life data only available for 2 years post-surgery, preventing assessment of long-term weight regain patterns that may differ between SAD and control groups.

“Response: Acknowledged as a significant limitation and emphasized the need for future studies with longer follow-up. Noted the difficulty of maintaining long-term follow-up in this population.”

Limits the ability to draw conclusions about long-term efficacy of bariatric surgery in the SAD population, particularly given external evidence suggesting potential divergence in weight trajectories beyond 2 years.

Authors response: We agree with the reviewer that missing data over long-term follow-up remains a limitation. Long-term weight effects, including recurrent weight gain, is highly relevant. Unfortunately, it remains very challenging to attain long-term weight data in general after bariatric surgery, and for patients with SAD in particular. For the current study, we can only acknowledge this as a limitation and have presented this in the current manuscript.

3.: The reviewer suggested exploring whether behavioral factors (e.g., anxiety-related differences in care-seeking behavior, lower threshold for reporting symptoms, or differential monitoring intensity) might contribute to the higher rate of non-serious complications in the SAD group.

“Response: Discussed biological mechanisms (SSRI effects on bleeding/clotting) but provided limited discussion of behavioral factors.”

Partially addressed with focus on biological rather than behavioral mechanisms.

The mechanistic understanding could be enhanced by considering both biological and behavioral pathways, but this does not fundamentally alter the study’s findings or conclusions.

Authors response: Thank you! We agree that the likely mechanism is a combination of biological and behavioral factors. The discussion has been expanded to further discuss differences in health seeking patterns, experiences of reassurance as well as somatic symptoms related to anxiety which may be of relevance in the postoperative setting.

Reviewer #2: Dear Authors,

In my opinion, you have addressed all the comments appropriately. I am satisfied with the manuscript in its current form, acknowledging both its strengths and limitations. I believe the insights it provides are valuable to the metabolic and bariatric surgery community. I look forward to seeing more interesting results from your database in the future.

Authors response: Thank you!

Reviewer #3: Dear Authors,

Thanks for addressing my comments. I think your manuscript can be publishable in PLOS One journal.

Authors response: Thank you!

Reviewer #4: All my comments have been addressed.

However, I noticed an inconsistent use of the abbreviation HRQoL, it is sometimes used and sometimes spelled out.

In the figure legend for Figure 1, p < 0.001 for all is stated twice. It should be sufficient to mention it once.

Authors response: Thank you! The legends have been revised as suggested.

---

## [Decision Letter · Decision Letter 2]

4 Jan 2026

Metabolic and bariatric surgery among patients with social anxiety disorder, a matched cohort study.

PONE-D-25-31803R2

Dear Dr. Stenberg,

We’re pleased to inform you that your manuscript has been judged scientifically suitable for publication and will be formally accepted for publication once it meets all outstanding technical requirements.

Kind regards,

M Saad Saumtally

Academic Editor

PLOS One

Additional Editor Comments (optional):

Thank you, and congratulations on the acceptance of your manuscript.

Reviewers' comments:

Reviewer's Responses to Questions

**Comments to the Author**

Reviewer #1: All comments have been addressed

Reviewer #3: All comments have been addressed

2. Is the manuscript technically sound, and do the data support the conclusions?

Reviewer #1: Yes

Reviewer #3: Yes

3. Has the statistical analysis been performed appropriately and rigorously?

Reviewer #1: Yes

Reviewer #3: Yes

4. Have the authors made all data underlying the findings in their manuscript fully available?

Reviewer #1: Yes

Reviewer #3: Yes

5. Is the manuscript presented in an intelligible fashion and written in standard English?

Reviewer #1: Yes

Reviewer #3: Yes

Reviewer #1: The authors did their best to address the reviewers' comments. I recommend acceptance of the manuscript as is.

Reviewer #3: Dear Authors,

Thank you for addressing all my comments. In my opinion your edited paper is suitable for publication.

**Do you want your identity to be public for this peer review?** For information about this choice, including consent withdrawal, please see our Privacy Policy

Reviewer #1: **Yes:** Mohamed Hany

Reviewer #3: **Yes:** Mohammad Kermansaravi

---

## [Editor Report · Acceptance letter]

PONE-D-25-31803R2

PLOS One

Dear Dr. Stenberg,

I'm pleased to inform you that your manuscript has been deemed suitable for publication in PLOS One. Congratulations! Your manuscript is now being handed over to our production team.

Kind regards,

on behalf of

Dr. M Saad Saumtally

Academic Editor

PLOS One